# Comment on "Magnetosensitive neurons mediate geomagnetic orientation in *Caenorhabditis elegans*"

Lukas Landler, Simon Nimpf, Tobias Hochstoeger, Gregory C Nordmann, Artemis Papadaki-Anastasopoulou, David A Keays*

Research Institute of Molecular Pathology, Vienna, Austria

**Abstract** A diverse array of species on the planet employ the Earth's magnetic field as a navigational aid. As the majority of these animals are migratory, their utility to interrogate the molecular and cellular basis of the magnetic sense is limited. Vidal-Gadea and colleagues recently argued that the worm *Caenorhabditis elegans* possesses a magnetic sense that guides their vertical movement in soil. In making this claim, they relied on three different behavioral assays that involved magnetic stimuli. Here, we set out to replicate their results employing blinded protocols and double wrapped coils that control for heat generation. We find no evidence supporting the existence of a magnetic sense in *C. elegans*. We further show that the Vidal-Gadea hypothesis is problematic as the adoption of a correction angle and a fixed trajectory relative to the Earth's magnetic inclination does not necessarily result in vertical movement.
DOI: https://doi.org/10.7554/eLife.30187.001

## Introduction

The ability to sense the Earth's magnetic field is a widespread sensory faculty in the animal kingdom (*Wiltschko and Wiltschko, 2012*). Magnetic sensation has been shown in migratory birds (*Zapka et al., 2009*), mole rats (*Nemec et al., 2001*), pigeons (*Keeton, 1971*; *Lefeldt et al., 2014*; *Mora et al., 2004*), and turtles (*Lohmann et al., 2004*). While behavioral evidence supporting the existence of a magnetic sense is strong, the underlying sensory mechanisms and neuronal circuitry that transduce and integrate magnetic information are largely unknown. A major impediment to progress in the field is the lack of genetic and molecular tools in magnetosensitive species. One such model system could be the nematode *Caenorhabditis elegans*, which has proved to be a powerful tool to explore a wide variety of senses. It has been claimed by *Vidal-Gadea et al. (2015)* that *C. elegans* possess a magnetic sense which can easily be exploited for mechanistic investigation (see also *Bainbridge et al., 2016*). They argue that *C. elegans* possess a magnetic sense that is employed for vertical orientation, worms adopting a correction angle relative to the inclination of the Earth's magnetic field. This conclusion was based on results from three assays which they developed: (1) a 'vertical burrowing assay'; (2) a 'horizontal plate assay'; and (3) a 'magnetotaxis assay'. Here, we set out to replicate the aforementioned behavioral assays, adopting several critical controls that were absent in the original study.

## Results

### Benzaldehyde control experiment

We established a positive control for our experiments employing the odorant benzaldehyde. It has been shown that if worms are placed in the center of a petri-dish and given the choice between 1% benzaldehyde and 100% ethanol they are attracted to the benzaldehyde. Conversely, if worms are

*For correspondence:
keays@imp.ac.at

Competing interests: The authors declare that no competing interests exist.

pre-exposed to 100% benzaldehyde their preference is disrupted (*Nuttley et al., 2001*). Employing blinded protocols, we found that worms preferred 1% benzaldehyde (n = 11, p<0.005, Wilcoxon signed rank test), which was lost when pre-exposed to 100% benzaldehyde (*Figure 1A–B*). These results show that we are able to replicate published *C. elegans* chemotaxis experiments in our laboratory.

## Infrastructure and double wrapped coils

To perform the magnetic experiments described by Vidal-Gadea and colleagues we built the necessary infrastructure to insure that our experiments were performed in a clean magnetic environment.

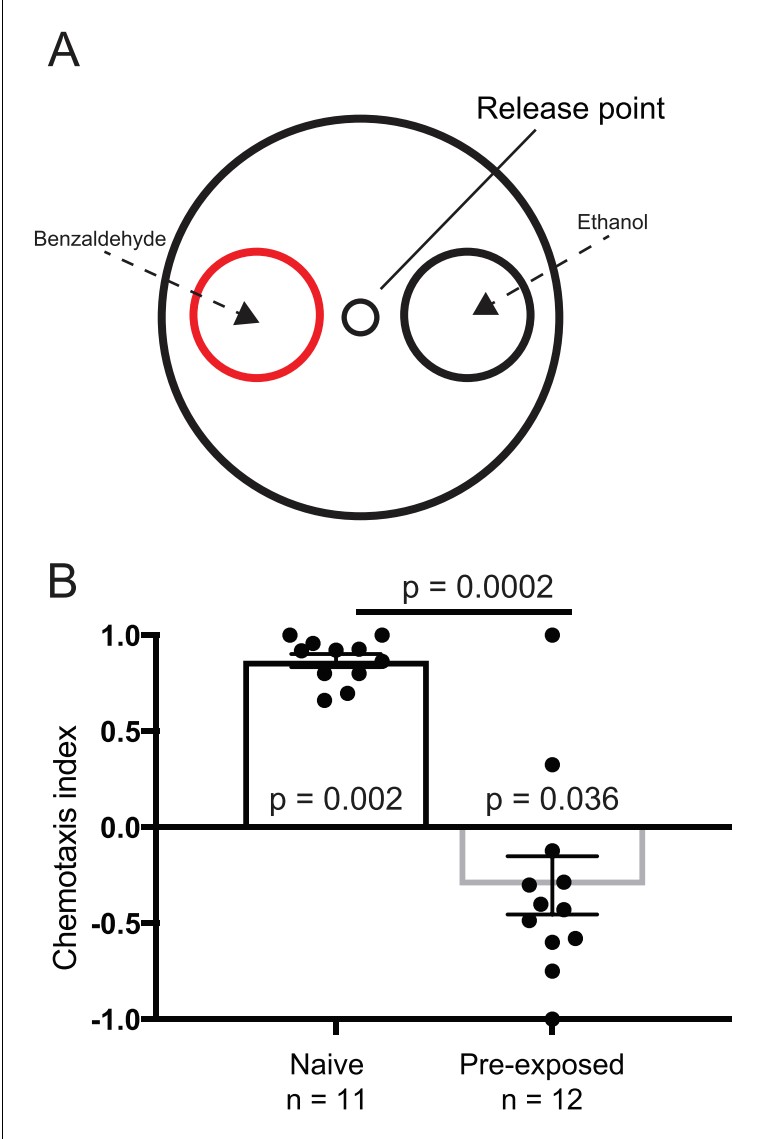

**Figure 1.** Benzaldehyde control experiment. (**A**) Experimental set up for the benzaldehyde-positive control experiments. Worms were placed at the release point and given a choice between 1% benzaldehyde in ethanol, or 100% ethanol. (**B**) Naive worms preferentially orientated toward the benzaldehyde (n = 11, p=0.002), and away from it if pre-exposed to benzaldehyde (n = 12, p=0.036). Each data point represents the result of one independent test plate.

DOI: https://doi.org/10.7554/eLife.30187.002

This consists of six double-wrapped Helmholtz coils, within a mu-metal shielded room that is surrounded by a Faraday cage (*Figure 2A–C*). Radio frequency contamination within this room is very low, with intensities below 0.1 nT between 0.1 to 10 MHz (see *Figure 2A–B*). This infrastructure is critical for applying magnetic stimuli in a controlled fashion (*Engels et al., 2014*).

### Vertical burrowing assay

In the first magnetic assay described by Vidal-Gadea, starved animals were injected into agar-filled plastic pipettes (*Figure 3A*). Worms were allowed to migrate overnight, and the number on each end of the tube were counted. In the absence of an external field the authors reported that animals preferentially migrated downwards, however, when exposed to an inverted Earth strength magnetic field worms migrated upwards. This preference was reversed in the case of fed animals. We repeated these experiments, but observed no effect of inverting the magnetic field on the burrowing index when the worms were starved (Mann-Whitney U-test, $n_1 = 38$, $n_2 = 40$, $U = 681$, n. s.) or fed (Mann-Whitney U-test, inclination down: $n_1 = 20$, $n_2 = 35$, $U = 300$, n. s.) (*Figure 3B*). The 95% confidence intervals for our experiments did not encompass the results Vidal-Gadea and colleagues

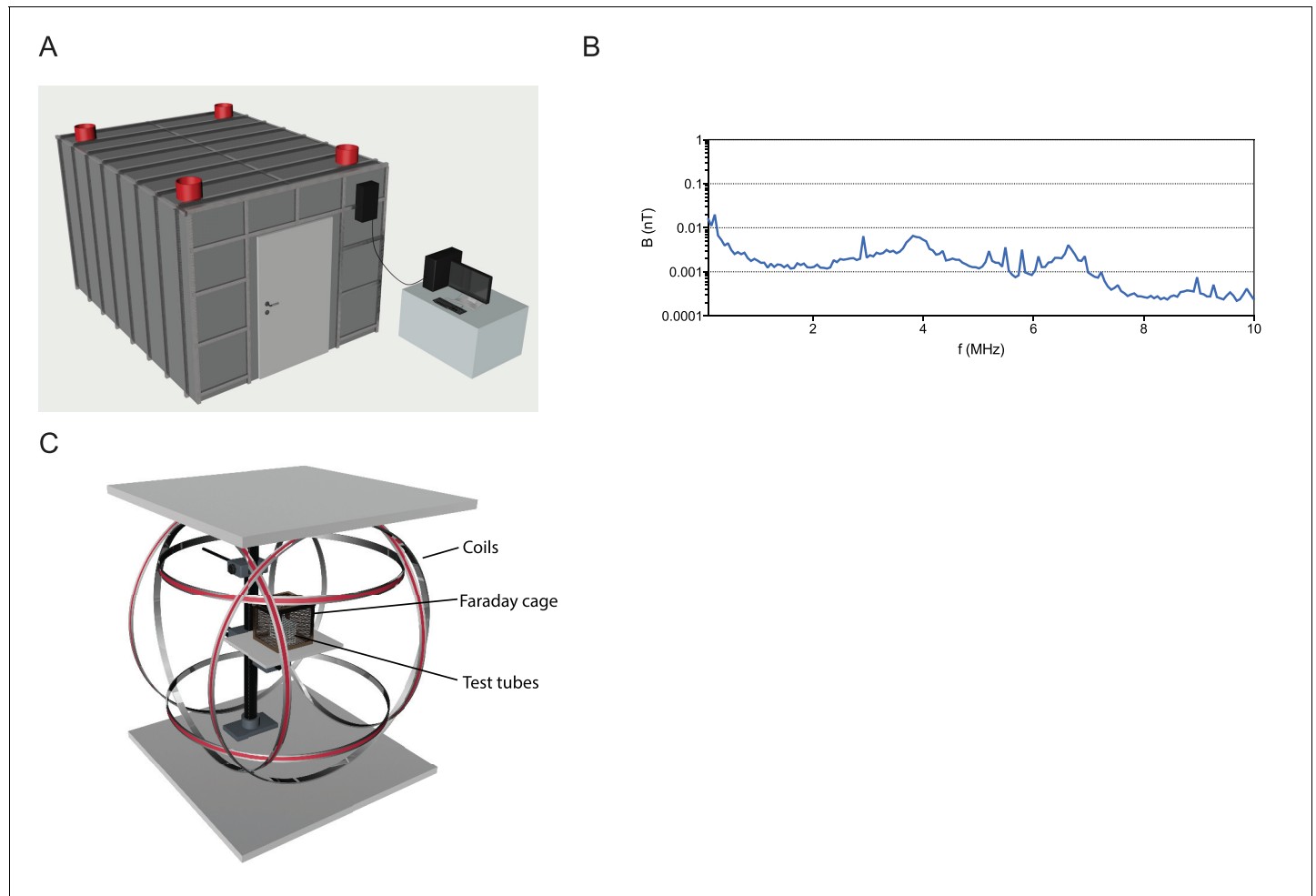

**Figure 2.** Infrastructure for magnetic experiments. (**A**) All experiments were performed within a mu-metal shielded room surrounded by a 5 mm aluminum Faraday cage. DC power sources and the computer driving the Helmholtz coils were located outside this shielded room, and cables into the room were filtered for radio frequencies. (**B**) Graph showing the radio-frequencies present in the shielded room between 0.1 to 10 MHz are below 0.1 nT, indicative of very low levels of radio frequency contamination. (**C**) Experimental setup for exposure of worms to magnetic fields. Three pairs of double-wrapped Helmholtz coils surround a plastic stage in the center. Worms were placed on this stage for the vertical burrowing, horizontal plate, and magnetotaxis assays. In the burrowing assay, we surrounded the tubes by an additional small Faraday cage.
DOI: https://doi.org/10.7554/eLife.30187.003

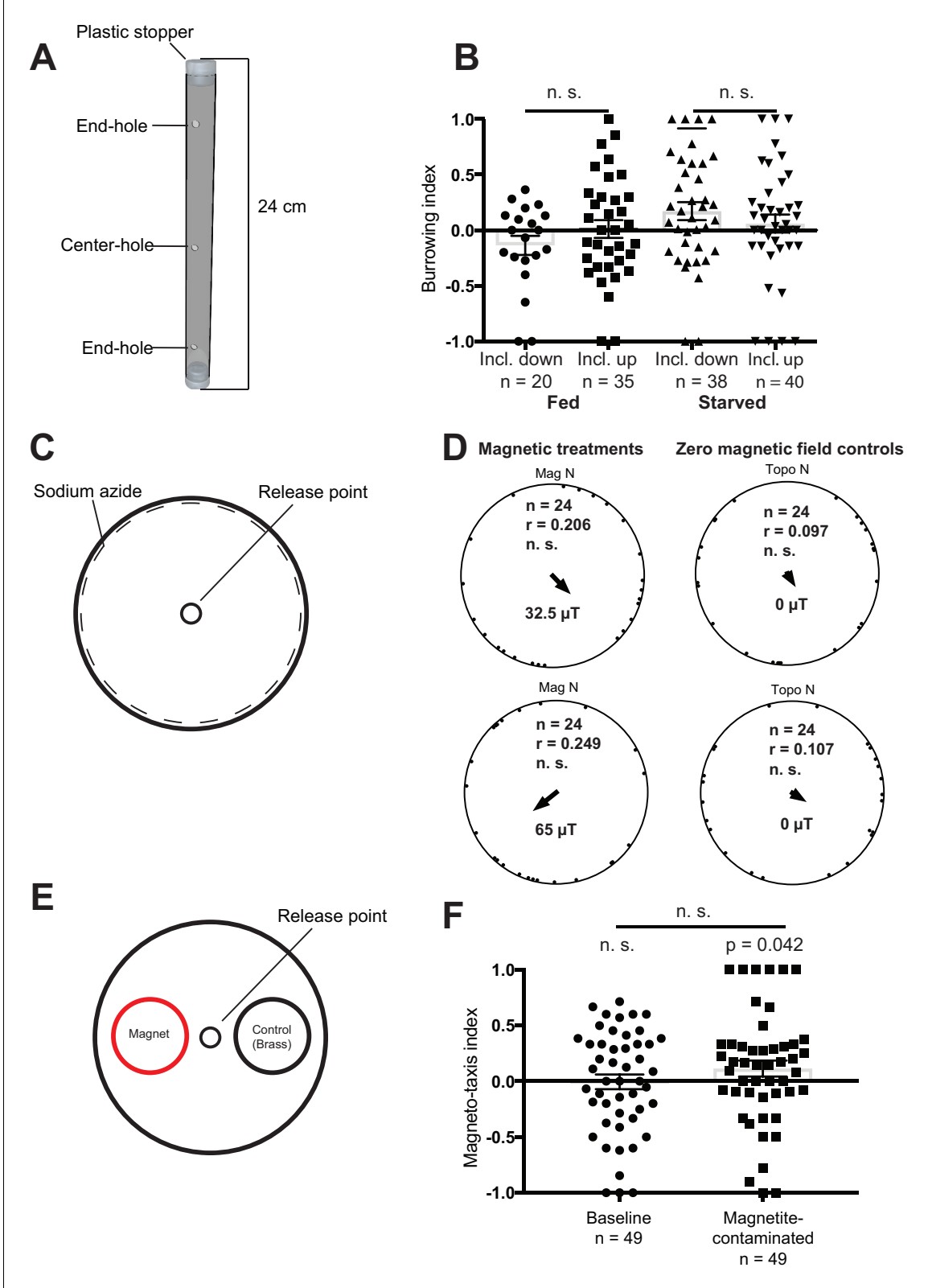

**Figure 3.** Magnetic assays and results. (A) Diagram showing the tubes employed for the vertical burrowing assay. Worms were injected in the center hole, and NaN$_3$ in the end-holes to immobilize them. Fed or starved worms were allowed to burrow overnight with the inclination of the magnetic field either up (59.16°) or down (−59.16°). At the conclusion of the test, the worms on either side (3 cm from the end hole) were counted and a preference index calculated. (B) Results for the vertical burrowing assay. We observed no significant difference in the burrowing index when the inclination of the

*Figure 3 continued on next page*

*Figure 3 continued*

magnetic field was inverted, whether the worms were fed or starved. (C) Set up for the horizontal plate assay. Worms were released in the center of the plate and allowed to move freely for 1 hr before the position and the direction of each worm relative to the center was recorded. Animals were tested in one of four magnetic directions (magnetic north pointing toward either topographic north, east, south, and west), with a field strength of 32.5 μT and 65 μT. Control experiments employed antiparallel currents resulting in a zero magnetic field. We calculated one mean orientation vector for each test plate by calculating the vector sum of all worms from this plate. (D) Results for the horizontal plate assay. We observed no directional preference when worms were exposed to either 32.5 μT or 65 μT magnetic stimuli. Each dot represents the mean worm direction for one plate, while the black arrow showing the direction and length (r) of the mean vector (radius of the circle is 1). Mag N indicates the normalized magnetic north and Topo N the topographic north. (E) Set up for the magneto-taxis assay. Worms were released in the center of a testing plate and could choose between two 3.5 cm diameter circles (goal areas) with a strong magnet (0.29 T) or a brass control underneath. Worms in each of the goal areas were counted and a preference index calculated. (F) We observed no preference for the area above the magnet, unless worms were fed bacteria contaminated with magnetite particles (p = 0.042, n = 49 plates). Error bars show standard error of the means.

DOI: https://doi.org/10.7554/eLife.30187.004

The following figure supplement is available for figure 3:

**Figure supplement 1.** Results of the burrowing assay performed on fed and starved worms in the absence of a magnetic field.

DOI: https://doi.org/10.7554/eLife.30187.005

obtained for the respective groups (see *Supplementary file 1*). Moreover, in the absence of a magnetic stimulus we found that the distribution of starved and fed worms was similar to data with an applied magentic field (*Figure 3—figure supplement 1*).

## Horizontal plate assay

In their second behavioral assay, Vidal-Gadea placed ≈50 worms in the center of an agar plate (*Figure 3C*). This plate was placed within a single wrapped Merritt coil system which permitted the generation of either null or horizontal magnetic fields of Earth strength intensity (either 32.5 μT or 65 μT). They reported that in the absence of magnetic stimuli worms displayed no directional preference, whereas in the presence of a horizontal field fed worms distributed in a biased direction 120° from north. We replicated these experiments, treating each plate as an experimental unit. Blind analysis of worm orientation revealed no effect on orientation behavior when applying a 32.5 μT stimulus (Rayleigh-test, r = 0.20, n = 24, n.s.) or a 65 μT stimulus (Rayleigh-test, r = 0.25, n = 24, n. s., *Figure 3D*). Nor did we observe any directional preference in our control experiments (32.5 μT: Rayleigh-test, r = 0.10, n = 24, n. s.; 65 μT: Rayleigh-test, r = 0.11, n = 24).

## Magnetotaxis assay

In their third behavior assay, worms were placed in the center of a horizontal agar plate between two different goal areas (*Figure 3E*). An extremely strong neodynium magnet generating a field up to 0.29 T (approximately 8000 times Earth strength), was placed beneath one of the goal areas. Vidal-Gadea reported that in the absence of this magnet worms were distributed evenly between the goal areas, however, if the magnet was present worms migrated toward it. We replicated their set up placing a strong neodynium magnet under one goal area, but added an equally size nonmagnetic brass control under the opposing goal area. We observed no preference for the goal area associated with the neodynium magnet (n = 49 plates, Wilcoxon signed rank test, V = 565, n.s., *Figure 3F*). The confidence interval did not include the results reported by Vidal-Gadea and colleagues (95% CI: −0.138 to 0.128). As false-positives in magnetoreception have been associated with contamination of biological material with exogenous iron we asked whether this might influence the behavior of worms (*Edelman et al., 2015*). We tested this by growing worms on agar plates spiked with magnetite particles, and repeated the magnetotaxis assay. We found a weak but significant preference for the goal area under which the magnet resided (Wilcoxon signed rank test, n = 49 plates, V = 670.5, p = 0.042, *Figure 3F*).

## Discussion

Why are our results different from those of Vidal-Gadea? We have gone to great lengths to employ the same protocols. We have used worms from the same source, we have employed the same neodymium magnets, we have used the same assay plates, and the same synchronization and starvation

protocols. There were, however, a number of important differences. First, we have used double wrapped coils for our experiments (*Kirschvink, 1992*). Our double wrapped coils (unlike single wrapped coils) allow the application of a magnetic stimulus without generating a change in temperature compared to the control condition. Heat is an issue when dealing with *C. elegans* as it is known that they can reliably detect temperature changes that are <0.1 ℃ (*Ramot et al., 2008*). Second, we used strict blinding procedures in all our assays, assuring an unbiased assessment of the worm responses. While Vidal-Gadea report blinding when comparing different genotypes, they do not report blinding to the magnetic condition. Third, we have applied the appropriate statistical methodology when analysing our data from the horizontal plate assay. Vidal-Gadea placed ≈50 worms on a plate treating each worm as a biological replicate. However, as worms tested on the same plate can interact with each other, they are not true independent biological replicates. The approach adopted by Vidale-Gadea is known as pseudoreplication, as it confuses the number of data points with the number of independent samples, increasing the probability of rejecting the null hypothesis whilst it is actually true (*Lazic, 2010*).

Moreover, there are a number of conceptual issues that undermine the assertion that *C. elegans* are magnetosensitive. First, the magnetotaxis assay relies on a permanent magnet that generates a field that is up to 8000 times Earth strength (0.29 T). At no time in its natural environment would *C. elegans* encounter such a strong field. An alternative explanation for this 'magnetotactic behavior' could be that exogenous iron particles attached to, or ingested by the worm, might, in the presence of an extremely large magnetic field influence the direction of locomotion by applying a force to surface mechanoreceptors.

More troubling is the underlying hypothesis that nematodes adopt a correction angle (α) relative to the inclination of the field to guide their vertical movement. Imagine a nematode is located in Cairo where the inclination of the Earth's magnetic vector is 44° 33'. To migrate vertically (i.e. 90°) it should adopt a correction angle of approximately 45° to the magnetic vector and maintain that trajectory (*Figure 4A*). Assuming that nematodes cannot distinguish up from down, the adoption of a fixed 45° angle from the inclination of the field is just as likely to result in horizontal movement (180°) as vertical translation (90°). This problem is exacerbated as the correction angle increases (e.g. 60°) (*Figure 4B*). In the best case scenario, worms could undertake random walks around a set angle (45°), that would result in a meandering descending trajectory, but with a large increase in path

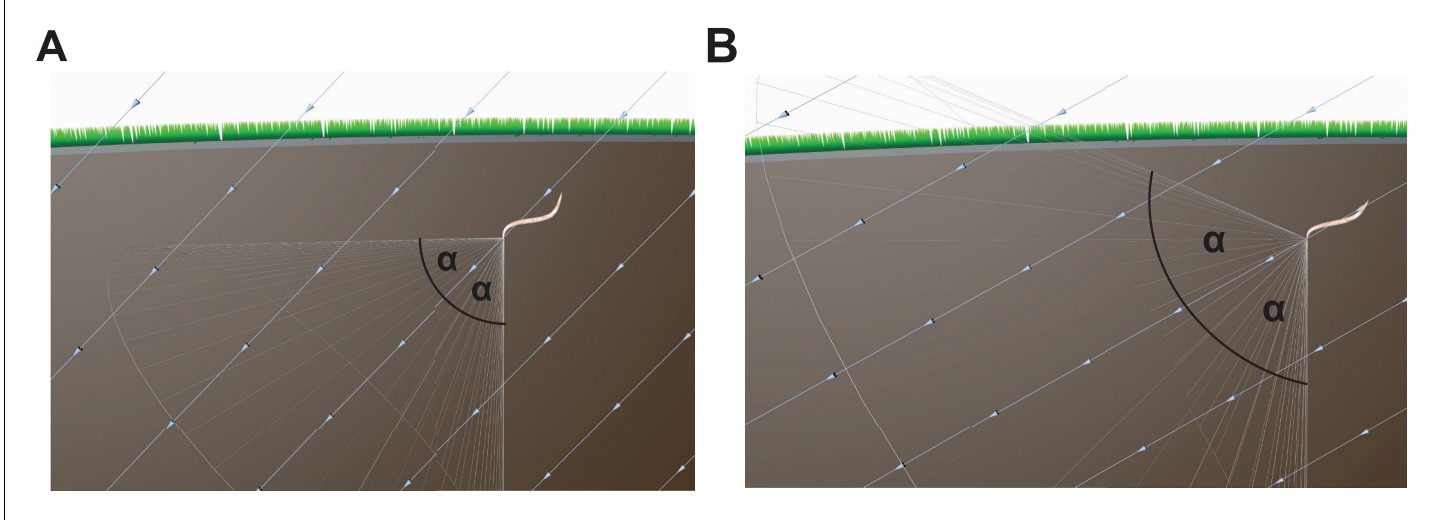

**Figure 4.** Conceptual issues with the Vidal-Gadea hypothesis. (**A**) The hypothesis advanced by Vidal-Gadea and colleagues argues that nematodes exploit the inclination of the Earth's magnetic field to guide vertical movement. They propose that nematodes adopt a correction angle (α, e.g. 45°) relative to the inclination of the field, which varies depending on the latitude. However, if the worms adopt such an angle and take a fixed trajectory this is as likely to result in a worm that travels horizontally as vertically. (**B**) As the latitude nears the equator the correction angle increases (e.g. 60°), and consequently a worm is just as likely to translate downwards, or at an oblique angle toward the Earth's surface. The light blue lines show the magnetic field vector.

DOI: https://doi.org/10.7554/eLife.30187.006

length. The concept proposed by Vidal-Gadea is only an efficient strategy if the worms are using the 'correction angle' in relation to an independent reference (i.e. gravity). However, if worms are able to distinguish up from down based on gravity, why would they rely on a magnetic field vector?

In conclusion, we were not able to replicate the findings of Vidal-Gadea and colleagues. We have made a number of arguments why this might be the case, but it is possible that our failure to replicate this work is due to a factor we are not aware of. However, it is pertinent to note that other attempts to elicit magnetoreceptive behavior in *C. elegans* have also been unsuccessful (*Njus et al., 2015*). Collectively, we conclude that *C. elegans* is not a suitable model system to understand the molecular basis of magnetoreception because (a) they lack a magnetic sense, or, (b) their magneto-tactic behaviour is not robust.

## Methods and materials

### Animals

Worms (N2 strain, received from Caenorhabditis Genetics Center) were maintained on the *Escherichia coli* strain OP50 as food. They were kept in incubators at constant dark conditions at 20 °C in an unmanipulated Earth-strength magnetic field (Vienna: field strength: 49 µT, inclination: 64°). For all assays, we used adult hermaphrodite worms that had not previously been starved. Worms were synchronized (bleached) before the tests to make sure animals of the same age were employed for behavioural analysis. Worms referred to as 'fed' were always tested within 10 mins of being removed from the culture plate. 'Starved' animals were kept in liquid Nematode Growth Media (NGM) for $\approx 30$ min.

### Chemotaxis experiments

For our chemotaxis experiments, we used 100 mm style petri dishes filled with 3% chemotaxis agar as test plates. Employing a template we marked each of the test plates with one center release point (see *Figure 3E*) and two smaller 'scoring' circles (diameter: 3.5 cm). Sodium azide (1.5 µl of 1 M) was applied to the center of each of the scoring circles to immobilize the worms (*Nuttley et al., 2001*). Worms were picked from the culture plates and collected in a small drop of NGM on a parafilm strip. In order to reduce bacterial contamination we carefully removed liquid containing bacteria and replaced it with new NGM. Worms were pipetted onto the center of the assay plate and 1 µl 1% benzaldehyde solution (in ethanol) was applied to one scoring circle and 1 µl 100% ethanol was applied to the other scoring circle. The plates were covered with aluminum foil and placed in the shielded room and left undisturbed for one hour. For our pre-exposure experiments a strip of parafilm with a 2 µl drop of 100% benzaldehyde was placed on the upper inside lid of a plate. After 90 min of pre-exposure the worms were tested as described above. For all chemotaxis experiments, we tested $\approx 50$ worms per test. A preference index (PI) was calculated by ascertaining the difference between the number of worms reaching the benzaldehyde decision circle (B) and the 100% ethanol decision circle (E) and divided it by the total number of worms scored, PI=(B-E)/(B) + (E).

### Magnetic coil set-up and magnetic shielding

For earth-strength magnetic field manipulations, we used a double wrapped custom built Helmholtz coil system (Serviciencia, S. L). The coils were located in the center of a 4.4 m (long) x 2.9 m (wide) x 2.3 m (high) shielded room. The diameter of coils were as follows: 1200 mm (Z-axis), 1254 mm (Y-axis) and 1310 mm (X-axis). The room was shielded against static magnetic fields by a 1 mm thick layer of Mu-metal and against oscillating electromagnetic fields by an aluminum layer (5 mm) (Magnetic Shielding). The 'Inclination down' setting as used in this study comprises a magnetic field vector with a 25 µT horizontal component, −42 µT vertical component and an inclination of −59.16°. The vertical component was inverted in the 'inclination up' treatment. Static magnetic fields were measured using a Three-axis Fluxgate Magnetometer (Bartington Instruments, UK). Radio frequencies were measured using an EMI test receiver (Rhode and Schwarz: MNr: E01180) and an active shielded loop antenna 6507 (EMCO: MNr: E0575). The receiver was put on MAXHOLD and measurements were taken for one min.

## Burrowing assay

We used 24 cm long tubes filled with 3% chemotaxis agar (see *Figure 3A*), each end was closed with a plastic stopper. The tubes contained three small holes (3 mm in diameter), one in the center and two 10 cm apart from the center hole on either side. During filling of the tubes great care was taken to avoid air bubbles at the ends of the tubes. Tubes with air bubbles were discarded. 1.5 µl of 1 M $NaN_3$ was added to each end-hole of a test tube and $\approx 50$ were injected into the center-hole (*Figure 3A*). The test tube was then covered with aluminum foil and placed upright in a holder. The holder was placed in the shielded room inside a smaller copper Faraday cage (*Figure 2C*). Tubes were left undisturbed overnight or alternatively over a day. At the conclusion of the test the tubes were removed from the room and worms on either side (3 cm from the end hole) were counted. The 'Inclination down' setting as used in this study comprises a magnetic field vector with a 25 µT horizontal component, −42 µT vertical component and an inclination of −59.16°. The vertical component was inverted in the 'inclination up' treatment. These magnetic conditions were identical to those employed by Vidal-Gadea. We calculated the burrowing index (BI) by dividing the difference between worms on either side of the plastic tube (A), (B) by the total number of scoring worms, BI= (A-B)/(A) + (B).

## Horizontal plate assay

Non-starved worms ($\approx 50$) were placed, with a droplet of NGM, on the center of a 100 mm style petri dish filled with 3% chemotaxis agar. Sodium azide (0.1 M, 20 µl) was applied to the rim of the plate to immobilize the worms once they reached it. Worms were released from the NGM droplet by removing the liquid with a tissue. The plate was then immediately placed in the center of the magnetic coils, described above, and covered with aluminum foil. Animals were tested in one of four magnetic directions (magnetic north pointing toward topographic north, east, south or west), with a field strength of 32.5 µT and 65 µT (close to the strength of the horizontal component of the Earth's magnetic field). In addition, we used two control conditions where the double wrapped coils were switched to antiparallel currents, which resulted in a zero magnetic field. We performed this control for the 32.5 µT and 65 µT field settings. Worms were allowed to move freely on the plate for 1 hr, then the position and the direction of each worm relative to the center was recorded. Magnetic field conditions were set by a person not involved in the analysis. Treatments and field conditions were revealed after all worms were counted and the angles measured.

## Magneto-taxis assay

We used 100 mm style petri dishes filled with 3% chemotaxis agar as test plates, marked with one center release point and two smaller 'scoring' circles. Sodium azide (1.5 µl of 1 M) was applied to the center of each of the scoring circles to immobilize the worms. We randomly placed a magnet (N42 Neodymium 3.5 cm diameter magnet 5 mm thick and nickel-plated) under one goal area, and a brass coin with identical dimensions as a control under the opposing goal area. The magnet was placed with the magnetic north pole pointing up in all tests. $\approx 50$ worms were placed in the central release point with a droplet of NGM. After the worms were released by removing the liquid the plate was covered quickly with aluminum foil and placed in the shielded room. After 1 hr, the number of worms in each goal area were counted blind. It should be noted that Vidal-Gadea performed this experiment over 30 min; however, our pilot experiments showed that a longer time resulted in a higher percentage of worms in the goal areas. For our iron contamination experiments, the OP50 (in solution) was mixed thoroughly with magnetite to create a 1% magnetite/OP50 solution. Worms were then synchronized and grown on OP50 covered plates until they reached adulthood. Experiments were performed as described above. In order to avoid cross-contamination separate picks were used for the magnetite and non-magnetite trials. To calculate the preference index (PI) the number of worms on the magnetic side (M) were subtracted by the number of worms on the control side (C) and then divided by the total number of scoring worms, PI = (M - C)/ (M + C).

## Statistics

In all tests, the experimenter was blind to the particular treatment when counting the worms. In general, we counted all tests, and did not discount tests based on low numbers of scoring worms or similar criteria in order to have an unbiased result. However, in the rare cases where no worms scored,

the tests were excluded from further analysis. A one-tailed Wilcoxon one-sample test was used to test if worms preferred the benzaldehyde and the magnet. For the burrowing assay, we used a two-tailed Wilcoxon one-sample test to ascertain whether worms burrowing preference differed from zero. In order to compare groups we used a Mann–Whitney U test. All linear statistical tests were performed in R (*R core team, 2012*). The circular data from the horizontal plate assay were analyzed using Oriana 4. Worms tested together at the same time on the same plate can interact with each other and hence constitute non-independent samples. Therefore, we calculated one mean orientation vector for each test plate, by calculating the vector sum of all worms from this plate. The directions from the plates, relative to the magnetic field and a geographically fixed direction (door to the shielded room), were then tested for a significant unimodal orientation using the Rayleigh test. Full statistics are shown in *Supplementary file 1*.

## Acknowledgements

We wish to thank Boehringer Ingelheim for funding basic research at the Institute for Molecular Pathology. We also wish to acknowledge the IMP graphics department and the media kitchen. We are indebted to Manuel Zimmer, Annika Nichols, and Harris Kaplan for their assistance with the experiments described in this manuscript. We thank Andres Vidal-Gadea for hosting Lukas Lander for a week in his laboratory, and showing him how to perform the magnetic behavioral protocols. The *C. elegans* strain was provided by the CGC which is funded by NIH Office of Research Infrastructure Programs (P40 OD 010440).

## Additional information

### Funding

| Funder | Grant reference number | Author |
| --- | --- | --- |
| Austrian Science Fund | Y726 | Lukas Landler<br>Simon Nimpf<br>Tobias Hochstoeger<br>Gregory Nordmann<br>Artemis Papadaki-Anastasopoulou<br>David A Keays |
| Horizon 2020 Framework Programme | 336724 | Lukas Landler<br>Simon Nimpf<br>Tobias Hochstoeger<br>Gregory Nordmann<br>Artemis Papadaki-Anastasopoulou<br>David A Keays |

The funders had no role in study design, data collection and interpretation, or the decision to submit the work for publication.

### Author contributions

Lukas Landler, Conceptualization, Supervision, Funding acquisition, Writing—original draft, Project administration, Writing—review and editing; Simon Nimpf, Conceptualization, Data curation, Formal analysis, Investigation, Methodology, Writing—original draft, Writing—review and editing; Tobias Hochstoeger, Data curation, Methodology; Gregory C Nordmann, David A Keays, Methodology, Writing—review and editing; Artemis Papadaki-Anastasopoulou, Methodology

### Author ORCIDs

Lukas Landler http://orcid.org/0000-0002-5638-7924
Simon Nimpf http://orcid.org/0000-0001-6522-6172
Gregory C Nordmann https://orcid.org/0000-0001-8840-7777
David A Keays http://orcid.org/0000-0002-8343-8002

**Decision letter and Author response**
Decision letter https://doi.org/10.7554/eLife.30187.011
Author response https://doi.org/10.7554/eLife.30187.012

## Additional files

### Supplementary files

• Supplementary file 1. Detailed summary of statistics used for the chemotaxis as well as the magnetic assays.

DOI: https://doi.org/10.7554/eLife.30187.007

• Transparent reporting form

DOI: https://doi.org/10.7554/eLife.30187.008

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
