## [Decision Letter]

Thank you for submitting your article "Comment on "Magnetosensitive neurons mediate geomagnetic orientation in *Caenorhabditis elegans*" for consideration by *eLife*. Your article has been reviewed by three peer reviewers, and the evaluation has been overseen by a Reviewing Editor and Eve Marder as the Senior Editor. The following individual involved in review of your submission has agreed to reveal his identity: Markus Meister (Reviewer #1).

Summary:

There is general agreement that the biological basis for a magnetic sense is one of the last great frontiers in sensory biology. The vexed history of magnetodetection in many other systems supports the view that *C. elegans* is an excellent system for understanding magnetodetection at the cellular and molecular levels. That is why we welcome your challenge to the research of Vidal-Gadea and colleagues, as we all agree that it is important to determine the validity of the original study.

General comments:

1) Subsection “Vertical Burrowing Assay”: "We repeated these experiments, but observed no effect of inverting the magnetic field on the burrowing index when the worms were starved…(n.s.)": The authors' results are much stronger than stated here and their presentation should be revised. (1) The statement currently focuses on "no effect of inverting the field" but actually the observation is simply "no effect at all". There is no evidence for directional burrowing, i.e. a non-zero BI, under any of the conditions. (2) For making a negative claim it is not sufficient to merely state "no significant difference". You can get "no significant difference" simply by not experimenting hard enough. Absence of significance is not significance of absence. A better method (not the only one) is to state the 4 results as mean {plus minus} 95% confidence limits. Then interpret this by showing that the null hypothesis (BI = 0) is within the confidence limits in all 4 cases, and that the numbers reported by Vidal-Gadea Figure 1B and 2G are far outside the confidence limits.

2) “Horizontal Plate Assay”: Again, statistical significance is irrelevant for making your case. Please state the strength of the directionality effect with confidence limits. Test whether that is consistent with zero effect and whether it is consistent with the Vidal-Gadea’s claims. In this case the effects in Vidal-Gadea Figure 2 are rather weak and may well be consistent with your measurements.

3) “Magnetotaxis Assay”: Again, the p-value is not helpful. State the effect size with confidence limits and compare to the Null hypothesis and to Vidal-Gadea Figure 4.

4) Introduction section: "Magnetic sensation has been shown in…". The authors may want to express this more carefully, especially given their current pursuit. Several of the papers they cite here contain weaker behavioral evidence than the Vidal-Gadea 2015 report. Wu and Dickman 2012 contains strong data, but has never been replicated, not even in the PI's laboratory, and not for lack of trying.

5) Subsection “Infrastructure and double wrapped coils”: "Radio frequency contamination within this room is very low…" What hypothesis would lead one to be interested in radio frequency magnetic fields? If you want to reassure us about possible environmental confounds, I would be more interested in odor plumes for example.

6) Subsection “Horizontal Plate Assay”: "worms distributed in a biased direction 60° either side of the imposed vector.": Actually in Vidal-Gadea Figure 2, the direction is 120 degrees from North for fed worms, and 60 degrees from North for starved worms.

7) Subsection “Magnetotaxis Assay”: "was placed beneath one of the goal areas": Strictly speaking Vidal-Gadea placed the magnet "above" the goal area.

8) Discussion section, paragraph three: More generally, worms should with equal probability choose directions on a cone whose surface is 45 degrees relative to the magnetic axis. On average they will therefore travel in the direction of the magnetic axis, but slower by a factor of 1/sqrt(2). So if the goal is to descend vertically, this strategy will take 41% longer on average than simply following the field.

9) Materials and methods section: Define "OP50"?

10) Figure 1: Maybe state that each dot is an independent test involving >50 worms.

11) Figure 2 legend: Is Figure 2B a power spectrum? If so, please state and label vertical axis accordingly.

12) I understand the authors' argument regarding the use of large double-wrapped coils instead of the single-wrapped coils used by Vidal-Gadea et al. However, it would be valuable for the authors to repeat the Vidal-Gadea experiments exactly according to their protocols, to see if this difference (and presumably generated heat) is responsible for the different results. (While I agree with the authors' points about double blinding and biological replicates, neither of these factors seems likely to explain the very divergent experimental outcomes.)

13) On a similar note, in the acknowledgements, it is stated that one of the authors (LL) visited the Vidal-Gadea lab. Did LL try to replicate the magnetotaxis results when he was there? Or did this visit simply involve an observation of protocols? Again, this might address specifically why the authors obtained different results from those in the Vidal-Gadea paper. At minimum, the authors should provide more detail about what the visits involved.

14) While the data presented challenge the results published by Vidal-Gadea and colleagues, we strongly recommend more modest interpretation of the data, especially omission of the last sentence of the paper. The failure of the replication cannot automatically be used to disprove original study. History of magnetoreception research teaches us that replication of some experiments may be difficult. For instance, it has taken years to replicate independently original experiments showing magnetic compass orientation in birds.

15) One potential caveat of the study is that the worms were kept and tested in different magnetic conditions. If I am right, the worms were kept in normal (i.e., rather noisy) magnetic field in a laboratory and then tested in the magnetically shielded Mu-metal chamber. It has been shown repeatedly in birds that pre-exposure to the same magnetic intensity and light conditions may be necessary for efficient magnetic orientation. For the sake of reproducibility and critical assessment of study design, magnetic and light conditions to which animals were exposed before and during the experiments must be described in detail.

16) The authors indicate levels of radiofrequency fields in the shielded experimental room between 0.5 and 5 MHz. Because it has been shown that broadband radiofrequencies (0.1 – 10 MHz) and single frequency of 7 MHz disrupt magnetic compass orientation of birds (Ritz et al., 2004), much broader frequency range has to be characterized to exclude possible interference with radical-pair mechanism. Information concerning levels of ELMF before and during experiment shall also be given.

17) While iron-contamination from the laboratory may lead to false positive results, the way in which the experiment was done seems unrealistic. The worms were grown on 1% magnetite/OP50 bacteria solution. Such a strong contamination is highly unlikely under standard lab conditions. In this context, the claim made in the Abstract (we … demonstrates that iron-contamination from laboratory settings can result in false positive results) is inappropriate and must be reformulated.

18) In the Discussion, the authors claim: "Our large double wrapped coils allow the application of a magnetic stimulus without generating a change in temperature compared to the control condition. In contrast, the small single wrapped coils employed by Vidal-Gadea generate a temperature gradient." This is a misleading statement for two reasons. First, Vidal-Gadea and colleagues used Merrit coil of 1 m3. It is not small for experiments with *C. elegans* that are performed on 100 mm petri dishes. Second, Vidal-Gadea et al. reported some measurable temperature gradient between center and periphery of petri dishes. However, there was no significant difference in this temperature gradient between different magnetic conditions (see their Figure 2—figure supplement 1). Thus, they actually did not face problem that your double wrapped coil solves. On the other hand, double wrapped coil in both parallel and anti-parallel mode will produce some heat and therefore potentially create temperate gradient on plates. Indicate size of Helmholz coils used in the study and temperature measurements if available.

19) The authors discuss problems associated with utilization of the inclination of the Earth magnetic vector as a cue for vertical movement. While the inclination is not an unambiguous cue, I do not agree that an independent reference (e.g. gravity) is needed. The inclination of magnetic vector is good enough for "directional random walk", which may suffice navigational needs of *C. elegans*.

[Editors' note: further revisions were requested prior to acceptance, as described below.]

Thank you for resubmitting your work entitled "Comment on "Magnetosensitive neurons mediate geomagnetic orientation in *Caenorhabditis elegans*" for further consideration at *eLife*. Your revised article has been favorably evaluated by Eve Marder (Senior editor), a Reviewing editor, and three reviewers.

The manuscript has been improved but there are some remaining issues that need to be addressed before acceptance, as outlined below:

General Comments:

The authors have generally done a good job responding to reviewer comments. We agree with their view that it is important to report instances in which earlier work is difficult or impossible to replicate, and the reviewers share your skepticism about aspects of the Vidal-Gadea paper, particularly the idea that worms use the earth's magnetic field to orient vertically in the soil.

Subsection “Infrastructure and double wrapped coils” and point 5 of the original review. Please add a citation of Engels, 2014 or some other explanation why you are measuring RF fields.

Subsection “Vertical Burrowing Assay”: Could the authors show control measurements without a magnetic field, and test whether the variance in worm trajectories is greater with than without field. This would address the concern that the assay may have averaged over fed and starved worms with positive and negative magnetotaxis.

Subsection “Magnetotaxis Assay”: "under one goal area", point 7 of the original review. To the reader this appears as a departure from Vidal-Gadea's published protocol "above one goal area". Please add the explanation you give in the reply to reviewers.

Discussion section, paragraph two: As pointed out in Vidal-Gadea's rebuttal, this is true only if the magnet were to lie in the plane of the worms. In reality the magnet is somewhat below that plane and that asymmetry produces a horizontal component to the field. Also in Figure 4A the field lines are unrealistic, far exaggerating the vertical component. This should be corrected or, perhaps better, the panel and its associated argument could be omitted.

Discussion section, paragraph three: "spiraling descending trajectory": I don't think a random walk would lead to any kind of spiral. But regardless of the overall shape of the path, if the worm always travels at 45 degrees relative to the intended direction, its rate of progress in that direction will be only 71% of optimum.

The first paragraph of the Discussion seems to attribute the lack of replication to one of three specific defects in Vidal-Gadea et al's experimental design. Without actually testing single wrapped coils in their hands, the authors cannot conclude that this is the causative factor. Likewise, I think the behavioral assay (counting paralyzed worms in a circle) should be relatively resistant to observer bias, and Vidal-Gadea's results look different from the author's irrespective of the statistical test used. It is quite possible, even likely, that a different factor explains the different results, and this should be noted in the Discussion.

---

## [Author Response]

General comments:1) Subsection “Vertical Burrowing Assay”: "We repeated these experiments, but observed no effect of inverting the magnetic field on the burrowing index when the worms were starved…(n.s.)": The authors' results are much stronger than stated here and their presentation should be revised. (1) The statement currently focuses on "no effect of inverting the field" but actually the observation is simply "no effect at all". There is no evidence for directional burrowing, i.e. a non-zero BI, under any of the conditions. (2) For making a negative claim it is not sufficient to merely state "no significant difference". You can get "no significant difference" simply by not experimenting hard enough. Absence of significance is not significance of absence. A better method (not the only one) is to state the 4 results as mean {plus minus} 95% confidence limits. Then interpret this by showing that the null hypothesis (BI = 0) is within the confidence limits in all 4 cases, and that the numbers reported by Vidal-Gadea Figure 1B and 2G are far outside the confidence limits.

We have added the 95% confidence intervals to our statistical analysis for the burrowing index (See Supplementary file 1). In all cases the mean results obtained by Vidal-Gadea and colleagues were outside our 95% confidence intervals. We now state this in subsection “Vertical Burrowing Assay”. The null hypothesis (BI=0) fell within the confidence interval in all cases, with the exception of worms that are starved and presented with a downward inclination field. In this case the lower confidence interval is 0.006, reflecting the fact that starved worms in general tend to burrow upwards. A Mann-Whitney U-test test confirmed the absence of a magnetic effect.

2) “Horizontal Plate Assay”: Again, statistical significance is irrelevant for making your case. Please state the strength of the directionality effect with confidence limits. Test whether that is consistent with zero effect and whether it is consistent with the Vidal-Gadea claims. In this case the effects in Vidal-Gadea Figure 2 are rather weak and may well be consistent with your measurements.

In the case of circular statistics the use of confidence intervals when the data distribution is wide is problematic. For instance, when applying a 32.5 µT field our confidence interval varied from 57.8° to 213.3°representing 43% of the possible variation in the dataset. Accordingly the likelihood that another dataset falls within this confidence interval is high, and not informative. We have nonetheless added confidence intervals to Supplementary file 1, but think they should be interpreted cautiously.

3) “Magnetotaxis Assay”: Again, the p-value is not helpful. State the effect size with confidence limits and compare to the Null hypothesis and to Vidal-Gadea Figure 4.

We now present the confidence intervals for the magnetotaxis assay in Supplementary file 1, and in the main text. The results reported by Vidal-Gadea for the N2 strain fall outside these confidence intervals – which we now state in the text.

4) Introduction section: "Magnetic sensation has been shown in…". The authors may want to express this more carefully, especially given their current pursuit. Several of the papers they cite here contain weaker behavioral evidence than the Vidal-Gadea 2015 report. Wu and Dickman, 2012 contains strong data, but has never been replicated, not even in the PI's laboratory, and not for lack of trying.

We appreciate that the Wu and Dickman electrophysiological studies have yet to be independently replicated, however, we think there is good evidence for a magnetic sense in pigeons. To address the reviewers’ concerns we now cite (Lefeldt et al., 2014), (Mora et al., 2004), and (Keeton, 1971).

5) Subsection “Infrastructure and double wrapped coils”: "Radio frequency contamination within this room is very low…" What hypothesis would lead one to be interested in radio frequency magnetic fields? If you want to reassure us about possible environmental confounds, I would be more interested in odor plumes for example.

Radio frequency noise at nT levels has been shown to disrupt the magnetic compass of animals, e.g. in migratory robins (Engels et al., 2014) presumably because of an underlying radical pair mechanism. Therefore RF shielding and measurements are common and important practice in magnetic orientation experiments. Odor plumes are an unlikely confounding factor in our case as the experiments were done using closed plates, or sealed tubes, in an enclosed room.

6) Subsection “Horizontal Plate Assay”: "worms distributed in a biased direction 60° either side of the imposed vector.": Actually in Vidal-Gadea Figure 2, the direction is 120 degrees from North for fed worms, and 60 degrees from North for starved worms.

We have clarified this in the text of our paper. We now state that "in the presence of a horizontal field fed worms distributed in a biased direction 120° from North".

7) Subsection “Magnetotaxis Assay”: "was placed beneath one of the goal areas": Strictly speaking Vidal-Gadea placed the magnet "above" the goal area.

In the supplementary material of their manuscript Vidal-Gadea et al., 2015 state that the magnet was placed “above one of the circles”. In initial preliminary trials we also performed the experiments this way. However, when LL visited the Vidal-Gadea lab, he was informed that in reality they perform their experiments by putting the plates on top of the magnets. We therefore performed the experiment in this way.

8) Discussion section, paragraph three: More generally, worms should with equal probability choose directions on a cone whose surface is 45 degrees relative to the magnetic axis. On average they will therefore travel in the direction of the magnetic axis, but slower by a factor of 1/sqrt(2). So if the goal is to descend vertically, this strategy will take 41% longer on average than simply following the field.

We appreciate the reviewers’ comment. Having given this matter some thought we think this depends on whether or not an individual worm adopts a fixed direction with respect to the magnetic vector. If the worm employs a "random walking" approach we think the reviewer is correct and you could expect them to descend around that vector in a spiraling fashion but at a slower rate. Alternatively, if the animal adopts a single fixed heading and maintains that vector it could either be going horizontally or vertically. To address this issue we have added the following sentence to the manuscript "In the best case scenario worms could undertake random walks around a set angle (45°), that would result in a spiraling descending trajectory, but with a large increase in path length."

9) Materials and methods section: Define "OP50"?

OP50 is the standard bacterial lawn (*E. coli*) used to feed *C. elegans*. We have clarified this in the Materials and methods.

10) Figure 1: Maybe state that each dot is an independent test involving >50 worms.

We have stated this in the figure legend.

11) Figure 2 legend: Is Figure 2B a power spectrum? If so, please state and label vertical axis accordingly.

In our view the axes are labeled correctly showing the maximum strength of the magnetic field (B, in nT) on the vertical axis and the frequency in MHz on the horizontal axis.

12) I understand the authors' argument regarding the use of large double-wrapped coils instead of the single-wrapped coils used by Vidal-Gadea et al. However, it would be valuable for the authors to repeat the Vidal-Gadea experiments exactly according to their protocols, to see if this difference (and presumably generated heat) is responsible for the different results. (While I agree with the authors' points about double blinding and biological replicates, neither of these factors seems likely to explain the very divergent experimental outcomes.)

We understand the reviewers’ argument, however, our goal was to ascertain whether or not *C. elegans* are a useful model system to study magnetoreception. We think there is little to be gained from investing time and effort in replicating experiments with obvious design flaws. To do so would require us to construct a new set of coils and re-do an entire set of experiments. Moreover, even if we adopted this course it is unlikely we could build an exact replica of the coils employed by Vidal-Gadea.

13) On a similar note, in the acknowledgements, it is stated that one of the authors (LL) visited the Vidal-Gadea lab. Did LL try to replicate the magnetotaxis results when he was there? Or did this visit simply involve an observation of protocols? Again, this might address specifically why the authors obtained different results from those in the Vidal-Gadea paper. At minimum, the authors should provide more detail about what the visits involved.

LL visited the Vidal-Gadea lab for one week and performed magnetotaxis and burrowing tests during this week. The primary goal of this visit was to learn and observe the behavioural protocols developed by Vidal-Gadea. An analysis of the results obtained did not reveal a significant magnetic effect, but the n number was low, and the experiments were not performed in shielded conditions and not always blinded. We have added information regarding the visit to the acknowledgements.

14) While the data presented challenge the results published by Vidal-Gadea and colleagues, we strongly recommend more modest interpretation of the data, especially omission of the last sentence of the paper. The failure of the replication cannot automatically be used to disprove original study. History of magnetoreception research teaches us that replication of some experiments may be difficult. For instance, it has taken years to replicate independently original experiments showing magnetic compass orientation in birds.

We are acutely aware of the difficulty of behavioral experiments in the field magnetosensation. We think this needs to be balanced with the need for independent replication of key papers. We do not, as stated by the reviewer claim to have "disproved" the original study. Proof is an ellusive and difficult goal to obtain, requiring (in our view) multiple independent labs that perform experiments that stand the test of time.

Rather our argument is that worms are not a good model to study magnetoreception. We have clarified this by modifying the last sentence so it now reads "Collectively, we conclude that *C. elegans* is not a suitable model system to understand the molecular basis of magnetoreception because (a) they lack a magnetic sense, or, (b) their magnetotactic behaviour is not robust."

15) One potential caveat of the study is that the worms were kept and tested in different magnetic conditions. If I am right, the warms were kept in normal (i.e., rather noisy) magnetic field in a laboratory and then tested in the magnetically shielded Mu-metal chamber. It has been shown repeatedly in birds that pre-exposure to the same magnetic intensity and light conditions may be necessary for efficient magnetic orientation. For the sake of reproducibility and critical assessment of study design, magnetic and light conditions to which animals were exposed before and during the experiments must be described in detail.

As requested we have added details to the methodology. Specifically, worms were kept in complete darkness in incubators which were kept constant at 20°C in the undisturbed Earth’s magnetic field in Vienna (field strength: 49µT, inclination: 64°). All experiments were performed in darkness.

16) The authors indicate levels of radiofrequency fields in the shielded experimental room between 0.5 and 5 MHz. Because it has been shown that broadband radiofrequencies (0.1 – 10 MHz) and single frequency of 7 MHz disrupt magnetic compass orientation of birds (Ritz et al., 2004), much broader frequency range has to be characterized to exclude possible interference with radical-pair mechanism. Information concerning levels of ELMF before and during experiment shall also be given.

In light of the reviewers’ request we have now included an expanded spectrum showing the radiofrequency fields in our mu metal shielded room. The frequency range now extends from 0.1 MHz to 10MHz, with intensities still below 0.1nT. We did not obtain spectra during the experiments.

17) While iron-contamination from the laboratory may lead to false positive results, the way in which the experiment was done seems unrealistic. The worms were grown on 1% magnetite/OP50 bacteria solution. Such a strong contamination is highly unlikely under standard lab conditions. In this context, the claim made in the Abstract (we.. . demonstrates that iron-contamination from laboratory settings can result in false positive results) is inappropriate and must be reformulated.

We have removed this sentence from the Abstract.

18) In the Discussion, the authors claim: "Our large double wrapped coils allow the application of a magnetic stimulus without generating a change in temperature compared to the control condition. In contrast, the small single wrapped coils employed by Vidal-Gadea generate a temperature gradient." This is a misleading statement for two reasons. First, Vidal-Gadea and colleagues used Merrit coil of 1 m3. It is not small for experiments with C. elegans that are performed on 100 mm petri dishes. Second, Vidal-Gadea et al. reported some measurable temperature gradient between center and periphery of petri dishes. However, there was no significant difference in this temperature gradient between different magnetic conditions (see their Figure 2—figure supplement 1). Thus, they actually did not face problem that your double wrapped coil solves. On the other hand, double wrapped coil in both parallel and anti-parallel mode will produce some heat and therefore potentially create temperate gradient on plates. Indicate size of Helmholz coils used in the study and temperature measurements if available.

We removed the statement regarding the size of the coils employed by Vidal-Gadea and added our coil size to the methods. It is true that our double-wrapped coils may produce some heat. Critically, however, any heat generated will be identical when comparing antiparallel and parallel conditions as the exact same amount of current is flowing in both cases. This cannot be said for single wrapped coils.

19) The authors discuss problems associated with utilization of the inclination of the Earth magnetic vector as a cue for vertical movement. While the inclination is not an unambiguous cue, I do not agree that an independent reference (e.g. gravity) is needed. The inclination of magnetic vector is good enough for "directional random walk", which may suffice navigational needs of C. elegans.

We agree with the reviewer that the inclination of the magnetic field alone would be a sufficient cue to move up or down (e.g. magnetotactic bacteria). However, the hypothesis advanced by Vidal-Gadea is that the worms adopt a correction angle relative to this vector (e.g. 45 degrees). If a worm follows a straight path, adopting a fixed correction angle, it will not necessarily result in vertical translocation – it is just as likely to end up travelling horizontally. However, we do acknowledge that the adoption of a "random walking" approach around an inclination angle would result in a worm that spirals downward (with a greater path length). We have modified the manuscript accordingly to reflect this.

[Editors' note: further revisions were requested prior to acceptance, as described below.]

General Comments:The authors have generally done a good job responding to reviewer comments. We agree with their view that it is important to report instances in which earlier work is difficult or impossible to replicate, and the reviewers share your skepticism about aspects of the Vidal-Gadea paper, particularly the idea that worms use the earth's magnetic field to orient vertically in the soil.

We are thankful for the positive assessment of our work, and feel that the additional changes which we have done in this round of revisions have further improved our manuscript.

Subsection “Infrastructure and double wrapped coils” and point 5 of the original review. Please add a citation of Engels, 2014 or some other explanation why you are measuring RF fields.

We added the requested citation.

Subsection “Vertical Burrowing Assay”: Could the authors show control measurements without a magnetic field, and test whether the variance in worm trajectories is greater with than without field. This would address the concern that the assay may have averaged over fed and starved worms with positive and negative magnetotaxis.

We have added these data (See Figure 3—figure supplement 1). We find that in the absence of a magnetic field the variance in trajectories when comparing starved and fed worms is similar.

Subsection “Magnetotaxis Assay”: "under one goal area", point 7 of the original review. To the reader this appears as a departure from Vidal-Gadea's published protocol "above one goal area". Please add the explanation you give in the reply to reviewers.

When LL visited the Vidal-Gadea Lab he was specifically shown to put the magnets underneath the plates, therefore we adopted the same method.

Discussion section, paragraph two: As pointed out in Vidal-Gadea's rebuttal, this is true only if the magnet were to lie in the plane of the worms. In reality the magnet is somewhat below that plane and that asymmetry produces a horizontal component to the field. Also in Figure 4A the field lines are unrealistic, far exaggerating the vertical component. This should be corrected or, perhaps better, the panel and its associated argument could be omitted.

In light of the reviewers’ comments we have omitted the panel and associated argument.

Discussion section, paragraph three: "spiraling descending trajectory": I don't think a random walk would lead to any kind of spiral. But regardless of the overall shape of the path, if the worm always travels at 45 degrees relative to the intended direction, its rate of progress in that direction will be only 71% of optimum.

We have changed the wording on this paragraph indicating the worms would undertake a "meandering descending trajectory"

The first paragraph of the Discussion seems to attribute the lack of replication to one of three specific defects in Vidal-Gadea et al's experimental design. Without actually testing single wrapped coils in their hands, the authors cannot conclude that this is the causative factor. Likewise, I think the behavioral assay (counting paralyzed worms in a circle) should be relatively resistant to observer bias, and Vidal-Gadea's results look different from the author's irrespective of the statistical test used. It is quite possible, even likely, that a different factor explains the different results, and this should be noted in the Discussion.

We now state "In conclusion, we were not able to replicate the findings of Vidal-Gadea and colleagues. We have made a number of arguments why this might be the case, but it is possible that our failure to replicate this work is due to a factor we are not aware of."